# The Important Role of N6-methyladenosine RNA Modification in Non-Small Cell Lung Cancer

**DOI:** 10.3390/genes12030440

**Published:** 2021-03-19

**Authors:** Yue Cheng, Meiqi Wang, Junliang Zhou, Huanhuan Dong, Shuqing Wang, Hui Xu

**Affiliations:** Department of Clinical Laboratory, Harbin Medical University Cancer Hospital, 150 Haping Road, Harbin 150081, China; cy@hrbmu.edu.cn (Y.C.); Wangmeiqi@hrbmu.edu.cn (M.W.); Zhoujunliang@hrbmu.edu.cn (J.Z.); dhh@hrbmu.edu.cn (H.D.); wangshuqing@hrbmu.edu.cn (S.W.)

**Keywords:** N6-methyladenosine (m^6^A), methyltransferases, demethylases, m^6^A-binding proteins, NSCLC

## Abstract

N6-methyladenosine (m^6^A) is one of the most prevalent epigenetic modifications of eukaryotic RNA. The m^6^A modification is a dynamic and reversible process, regulated by three kinds of regulator, including m^6^A methyltransferases, demethylases and m^6^A-binding proteins, and this modification plays a vital role in many diseases, especially in cancers. Accumulated evidence has proven that this modification has a significant effect on cellular biological functions and cancer progression; however, little is known about the effects of the m^6^A modification in non-small cell lung cancer (NSCLC). In this review, we summarized how various m^6^A regulators modulate m^6^A RNA metabolism and demonstrated the effect of m^6^A modification on the progression and cellular biological functions of NSCLC. We also discussed how m^6^A modification affects the treatment, drug resistance, diagnosis and prognosis of NSCLC patients.

## 1. Introduction

Lung cancer is the most common type of cancer, with malignant tumors having a high incidence and mortality rate worldwide [1]. Non-small cell lung cancer (NSCLC) is the main type of lung cancer, accounting for about 80% of lung cancer cases [2]. Although targeted therapy and immunotherapy have made breakthroughs in NSCLC treatment [3], gene mutation and PD-1 expression remain obstacles in the treatment, and the five-year survival rate of NSCLC patients is still unsatisfying [4]. Therefore, it is important to explore the molecular mechanisms of applied prognostic biomarkers and therapeutic targets.

Epigenetic modifications are required for diverse biological activities in mammalians, and the role of RNA epigenetic modifications in gene expression regulation is rapidly becoming clearer [5,6]. Among these modifications, the N6-methyladenosine (m^6^A) modification is the most prevalent and abundant modification in eukaryotes, and has been studied extensively. m^6^A refers to the methylation of the sixth N of adenylated RNA/DNA, and was discovered in the early 1970s [7]. The m^6^A modification widely exists in the consensus sequence RRACH (where R: A or G, H: A, C or U) [8,9] and was not only enriched around stop codons, but also in the coding sequence (internal long exons) and the 3′-untranslated region (3′-UTR) [10]. In addition to the effect on messenger RNA (mRNA) translation, degradation, splicing, export and folding [11,12], m^6^A methylation also regulates the metabolism and functions of a variety of noncoding RNAs (ncRNAs), including long noncoding RNAs (lncRNAs), microRNAs (miRNAs) and circular RNAs (circRNAs) [12,13,14,15].

## 2. The Regulation of RNA m^6^A Modification

The m^6^A modification has three kinds of regulator: m^6^A methyltransferases (also called “writers”), demethylases (also known as “erasers”) and m^6^A-binding proteins (also called “readers”), which influence various m^6^A RNA metabolisms (Table 1).

The m^6^A modification is done using the “writers”, mainly including the methyltransferase complex (MTC), which is mainly composed of methyltransferase-like protein 3 (METTL3), METTL14 and Wilms tumor 1-associated protein (WTAP) [6]. METTL3 forms a stable heterodimer complex together with METTL14, in a 1:1 ratio [39]. The METTL3–METTL14 complex catalyzes the methyl transfer and promotes RNA substrate recognition [39,40,41]. WTAP is responsible for localizing the METTL3–METTL14 heterodimer to the nuclear speckles and enhancing their catalytic activity [17]. METTL16 can bind to various ncRNAs (U6 small nuclear RNA (snRNA), lncRNAs) and pre-mRNAs. RNA–binding motif protein 15 (RBM15) facilitates the recruitment of MTC to the target sites in a WTAP–dependent manner [42].

Three m^6^A demethylases have been reported to date, including fat mass and obesity-associated protein (FTO), alkB homolog 5 (AlKBH5) and ALKBH3 [43,44,45]. They all belong to the alkB family of non-heme iron- (II) and 2-ketoglutarate-dependent dioxygenases [46]. FTO is mainly associated with adipogenesis [47] and can bind to mRNAs, snRNAs and tRNAs as an m^6^A demethylase [48]. The FTO-mediated demethylation of internal m^6^A preferentially presents in cell nucleus [48]. In terms of the structure, the catalytic activity of FTO can be affected by the sequence and the tertiary structure of RNAs [49]. ALKBH5 is associated with infertility and normal spermatogenesis [21].

The m^6^A-binding proteins refer to the proteins that recognize and bind to the m^6^A modification on RNA and regulate translation, decay, splicing and nuclear export [23,25,28,50]. Recent studies [51,52,53] have shown that the YT521-B homology (YTH) domain, which recognizes the sites containing methylated adenines and binds to a short, degenerated, single-stranded RNA sequence, has an m^6^A motif within its hydrophobic pocket. In humans, YTHDC1 is a putative nucleus m^6^A-binding protein [23], while YTHDF1, YTHDF2, YTHDF3 and YTHDC2 are cytoplasmic m^6^A-binding proteins [54]. Insulin-like growth factor 2 mRNA-binding proteins (IGF2BPs), a family of m^6^A readers [55], bind directly to target mRNA through their hnRNP-K homology (KH) domains, which may be indispensable for recognizing and binding m^6^A [56]. IGF2BPs include IGF2BP1, IGF2BP2 and IGF2BP3. IGF2BP1 has the most conserved oncogenic potential among IGF2BPs [57]. The heterogeneous nuclear ribonucleoprotein (HNRNP) particles usually exist in the forms of tetramers. They include hnRNP (C1) _3_C2, hnRNP (A2) _3_B1 and hnRNP (A1) _3_B2 [58,59,60]. The hnRNP (C1) _3_C2 (hnRNP C) is a core component of hnRNP particles [61] and can recruit hnRNP particles to pre-mRNA [62]. In terms of structure, hnRNP C shows a preference for binding to uridine (U)-rich sequences [63]. A recent study [12] shows that the heterogeneous nuclear ribonucleoprotein G (HNRNPG) was also an m^6^A reader. Three subunits of the eukaryotic initiation factor 3 (eIF3) complex are correlated with cancer development—subunit eIF3d possesses cap-binding activity [64]; eIF3G is associated with cancer progression; eIF3h interacts with METTL3 [65,66].

m^6^A methyltransferases and m^6^A demethylases modulate the level of m^6^A in RNA, and RNA m^6^A modification significantly affects RNA splicing, transcription, nuclear export, translation, decay and stability under the action of m^6^A-binding proteins (Figure 1). It can be seen that m^6^A modification is a dynamic, reversible and continuous process. Furthermore, interaction between m^6^A regulators is common. For example, METTL3, METTL14 and WTAP compose a complex that promotes the localization of MTC at nuclear speckles [67] and METTL3 interacts with eIF3h to enhance mRNA translation [66]. YTHDF3 modulates the target mRNAs translation in synergy with YTHDF1 and regulates the mRNAs decay, cooperating with YTHDF2 [32]. Interestingly, the RNA-binding affinity of YTHDF1 and YTHDF3 is negatively correlated [31]. Three YTHDF proteins work together to induce the degradation of target m^6^A-mRNAs and one of them can fully or partially compensate for the effect of the others on target m^6^A-mRNAs degradation [68]. Therefore, it is believed that m^6^A modification regulates m^6^A RNA metabolism via the collective effect of m^6^A regulators.

## 3. Dual Effects of RNA with m^6^A Modification on Progression and Cellular Biological Functions of NSCLC

m^6^A modification of mRNAs and lncRNAs regulates cellular biological functions in NSCLC via the control of m^6^A regulators. Additionally, the m^6^A mRNA modification not only plays an oncogene role, but also an anti-oncogene role (Table 2 and Table 3). m^6^A mRNA modification plays these different roles in the progression and cellular biological function of NSCLC, likely due to different hepatic inflammatory microenvironments [69].

### 3.1. Effects of m^6^A Modification of mRNAs on NSCLC

#### 3.1.1. mRNA m^6^A Modification Generates an Oncogene Effect on NSCLC

##### Messenger RNA Yes-Associated Protein (mRNA YAP)

YAP is one of the main effectors of the Hippo pathway. It plays an oncogenic role in the development, progression and prognosis of NSCLC [86]. Jin et al. [70] found that METTL3 methylated YAP pre-mRNA acts as an m^6^A methyltransferase to enhance YAP mRNA translation and expression in NSCLC cells. The increase in YAP m^6^A modification mediated by METTL3 induced resistance to cisplatin (cis-diamminedichloroplatinum, DDP) and metastasis in NSCLC. The mRNA acts as a target for m^6^A demethylase ALKBH5, which decreased YAP activity by regulating miR-107/LATS2 to inhibit NSCLC tumor growth and metastasis [75]. In addition, they found that YTHDF1, YTHDF2 and YTHDF3 all bound to YAP pre-mRNA, and recognized the m^6^A modification to regulate mRNA YAP expression. YTHDF3 recognized m^6^A modifications initiated by METTL3 and upregulated YAP expression to promote cellular growth, invasion and migration in NSCLC [70]. More importantly, YTHDF1 promoted mRNA translation to enhance the cellular ability of growth, invasion and epithelial–mesenchymal transition (EMT) in NSCLC [75].

##### Messenger RNA Enhancer of Zeste Homolog 2 (mRNA EZH2)

Histone methyltransferase EZH2 is the catalytic subunit of the polycomb repressive complex 2 (PRC2) and is known to participate in the histone deacetylase (HDAC) process [87]. METTL3-mediated m^6^A modification of EZH2 mRNA increased migration and invasion level in NSCLC, and Simvastatin generated anti-cancer effects by downregulating METTL3 expression [71].

##### mRNA JUNB

JUNB is an EMT-related transcription factor (TF) and regulates the ability of cellular EMT [88]. JUNB mRNA stability was decreased by reducing METTL3-directed m^6^A RNA modification in NSCLC cell lines. Elevated JUNB expression increased the TGF-β-induced migratory activity in NSCLC [72].

##### Messenger RNA 6-Phosphogluconate Dehydrogenase (mRNA 6PGD)

The occurrence of aerobic glycolysis is increased in cancer cells, which is known as the Warburg effect. As the third enzyme in the oxidative pentose phosphate pathway (PPP), 6PGD is vital in the process of regulating cancer cell metabolism [89,90]. Sheng et al. [81] reported that mRNA 6PGD knockdown inhibited the proliferation of NSCLC cell lines, and mRNA overexpression reversed the tumor cell growth inhibition mediated by YTHDF2 knockdown. In their work, YTHDF2 recognized that the m^6^A modification sites on the 6PGD mRNA 3′ UTR and YTHDF2 facilitate NSCLC cells growth and proliferation in an m^6^A-dependent way, via inducing 6PGD protein expression. Undoubtedly, the m^6^A methylation of mRNA 6PGD generates an oncogene effect via YTHDF2 in NSCLC.

##### mRNA MUC3A

The human intestinal mucin gene MUC3A encodes membrane-bound glycoprotein and is known to be an oncogene [91]. There is a low methylation level in MUC3A-high cell lines in colon adenocarcinoma [92]. Zhao et al. [74] found that the mRNA and m^6^A level of MUC3A mediated by vir-like m6A methyltransferase associated (VIRMA, also termed KIAA1429) had a positive correlation with the KIAA1429 level, which presented an opposing result to the previous study. In addition, overexpressing MUC3A induced cell proliferation, migration and invasion in lung adenocarcinoma (LUAD).

##### Messenger RNA Ubiquitin-Conjugating Enzyme E2C (mRNA UBE2C)

UBE2C encodes a class of ubiquitin-conjugating enzymes that are involved in ubiquitin-mediated protein degradation [93] and associated with the prognosis of stage I NSCLC patients [94]. The m^6^A and expression level of mRNA UBE2C was decreased by ALKBH5. Downregulation of the mRNA activated the expression of the autophagy markers ATG3 and LC3B and reduced cell proliferation, clonogenicity, and the invasive growth of NSCLC [76]. It can be seen that m^6^A modification is involved in deregulation of the UBE2C–autophagy repression axis.

#### 3.1.2. m^6^A Modification of mRNAs Generates an Anti-Oncogene Effect on NSCLC

##### mRNA YAP

The m^6^A modification on mRNA YAP not only generates an oncogene effect, but also anti-oncogene effect in NSCLC. Research [75] has confirmed that m^6^A modification generated an anti-oncogene effect via YTHDF2-mediated degradation of YAP by weakening the cellular abilities of growth, invasion and EMT in NSCLC.

##### Messenger RNA Forkhead Box M1 (mRNA FOXM1)

A study has shown that mRNA FOXM1 is a transcription factor that enhances cellular proliferative capacity in lung cancer [95]. ALKBH5 enhances the translational efficiency of mRNA FOXM1 and increases FOXM1 protein level by decreasing the FOXM1 mRNA m^6^A level. Increasing FOXM1 expression strengthens the abilities of cellular proliferation and invasion in LUAD under intermittent hypoxia (IH) [77].

##### Messenger RNA Tissue Inhibitor of Metalloproteinase 3 (mRNA TIMP3)

TIMP3 is an accepted tumor suppressor gene and can inhibit tumor invasion and metastasis [96,97]. ALKBH5 decreased the mRNA stability by reducing the m^6^A level to promote the malignant biological properties of NSCLC cells [78].

##### Messenger RNA Ubiquitin-Specific Protease-7 (mRNA USP7)

The deubiquitylating enzyme USP7 was previously characterized as a tumor suppressor gene, but was recently proven to promote cell proliferation in NSCLC [98,99]. The m^6^A level of mRNA USP7 was decreased by m^6^A demethylase FTO, which promoted NSCLC cell growth [79].

##### Messenger RNA Myeloid Zinc Finger 1 (mRNA MZF1)

MZF1, a member of the SCAN-Zinc Finger (SCAN-ZF) transcription factor family, may promote tumor progression and cause poor prognosis of the patients in LUAD [100]. The mRNA level of MZF1 was remarkably reduced, and the m^6^A modification of the mRNA transcripts was added by genetic silencing of the expression of FTO in lung squamous cell carcinoma (LUSC) cell lines. Forced expression of MZF1 partially reversed the suppressive effect on the viability and invasion of LUAC cell lines mediated by FTO knockdown [80]. Generally speaking, the m^6^A modification of mRNA MZF1 generates an anti-oncogene effect via FTO in LUSC.

##### Messenger RNA Solute Carrier 7A11 (mRNA SLC7A11)

Downregulating the expression of SLC7A11 inhibits cellular proliferation and colony formation in lung cancer [101]. The m^6^A modification destabilized SLC7A11 mRNA and accelerated mRNA decay after recognition by YTHDC2 in LUAD, which impaired cystine uptake. Significantly, this provides the theoretical basis for system X _C_^−^ -targeting therapy for LUAD [82].

##### mRNA NOTCH1

Some studies indicated that the Notch pathway works as an oncogene by downregulating the mRNA-suppressed NSCLC cell growth and invasion [102,103]. An mRNA degradation experiment in H1299 cells demonstrated that mRNA NOTCH1 stability can be promoted by circNOTCH1 [73]. For the purpose of exploring the effect of m^6^A modification on mRNA NOTCH1 stability, Shen et al. [73] performed RNA interaction–precipitation (RIP), which showed that NOTCH1 mRNA level was much lower in the control group than that in the shcircNOTCH1 group. These indicated that m^6^A modification related to METTL14 on NOTCH1 mRNA decreases mRNA stability in NSCLC cells. As a result, there is a chance that the m^6^A modification of NOTCH1 mRNA may generate an anti-oncogene effect via METTL14 in NSCLC.

### 3.2. Effects of m^6^A Modification of ncRNAs on NSCLC

#### 3.2.1. Long Non-Coding RNA Lung Adenocarcinoma Transcript 1 (lncRNA MALAT1)

lncRNA MALAT1 is up-regulated in NSCLC tissues and silencing of its expression represses cell proliferation, invasion and promotes apoptosis [104]. In addition, METTL3 increased the RNA expression level of the lncRNA in an m^6^A-dependent manner. In addition, YTHDF3 recognized the m^6^A modification sites on the lncRNA and enhanced the lncRNA stability in NSCLC cells [70].

#### 3.2.2. Long Non-Coding RNA FEZ Family Zinc Finger 1 Antisense RNA 1 (lncRNA FEZF1-AS1)

There are seven m^6^A-modified sites on lncRNA FEZF1-AS1. Knocking down lncRNA FEZF1-AS1 inhibits the proliferation, migration and invasion abilities of NSCLC cell lines [85]. Meanwhile, silencing METTL3, METTL14, YTHDF1 and YTHDF2 decreased the expression level of the lncRNA [85]. Therefore, it is possible that the m^6^A methylation of lncRNA FEZF1-AS1 plays an “oncogene” role via METTL3, METTL14, YTHDF1 and YTHDF2 in NSCLC.

#### 3.2.3. Long Non-Coding RNA ABHD11 Antisense RNA 1 (lncRNA ABHD11-AS1)

It has been proven that lncRNA ABHD11-AS1 contributes to many kinds of cancer tumorigenesis [105,106,107]. In addition, Xue et al. [83] found that there were modification sites on METTL3 that match the m^6^A modification of the lncRNA. METTL3 overexpression increased the m^6^A level of lncRNA ABHD11-AS1 and strengthened the stability of the lncRNA. lncRNA ABHD11-AS1 also promoted the Warburg effect of NSCLC via the EZH2/KLF4 axis.

#### 3.2.4. miR-143-3p

miR-143-3p induces EMT of NSCLC cells and is correlated with the progression of lung cancer. m^6^A modification can facilitate the splicing of precursor miR-143-3p and overexpression of METTL3 can increase the expression of miR-143-3p in NSCLC cells. Then, miR-143-3p promotes the brain metastasis of NSCLC in the condition of the regulation of vasohibin-1 (VASH1)/vascular endothelial growth factor A (VEGFA) axis [84].

There are other m^6^A modifications of ncRNAs that influence the progression and cellular biological function of NSCLC, which have yet to be uncovered. For instance, miR-376c was carried by endothelial cells (ECs)-secreted extracellular vesicles (EVs). Previous research has concluded that miR-376c overexpression could generate an inhibitory effect on Wnt signaling, NSCLC cell growth and invasion [108]. The overexpression of YTHDF1 could attenuate this effect by targeting miR-376c [109]. Interacting with heterogeneous nuclear ribonucleoprotein A2/B1 (HNRNPA2B1), the upregulation of LINC01234 mediated by c-Myc promotes the processing of primary miR-106b-5p. miR-106b-5p enhances NSCLC cell growth by inhibiting cryptochrome 2 (CRY2) expression and upregulating c-Myc. This forms a positive-feedback loop that is participated in by the m^6^A reader and miR-106b-5p and promotes NSCLC cell growth [110]. In addition, derived from the NADH: ubiquinone oxidoreductase subunit B2 (NDUFB2), circNDUFB2 is significantly downregulated in NSCLC and negatively correlated with malignant features of NSCLC. It serves as a scaffold to bridge TRIM25 and IGF2BPs and forms the TRIM25/circNDUFB2/IGF2BPs ternary complex, which accelerates IGF2BP ubiquitination and degradation. Interestingly, the acceleration effect is enhanced by m^6^A circNDUFB2 [111]. The effect of m^6^A modification on miR-106b-5p, miR-376c and circNDUFB2 to NSCLC remains to be explored.

## 4. Roles of m^6^A Modification on Treatment, Drug Resistance, Diagnosis and Prognosis of NSCLC Patients

m^6^A modification-related treatments have received widespread attention, such as anti-inflammatory cytokine interleukin 37 (IL-37), FTO and YTHDF1. IL-37 treatment inhibits the proliferation ability of LUAD cells via regulation of RNA methylation. Meanwhile, overexpression of IL-37 decreased the expression of ALKBH5. This means that the mechanism of IL-37 used for treating NSCLC patients may be associated with m^6^A methylation [112]. Although the emergence of targeted therapy provides a new direction for the treatment of NSCLC, drug resistance has become a major obstacle during the treatment. Some research has indicated that m^6^A modification enhanced or attenuated the drug-resistance of NSCLC patients, such as resistance to gefitinib, afatinib and DDP. m^6^A modification decreased the gefitinib-resistance (GR) of NSCLC cells via the FTO/YTHDF2/ABCC10 axis [113]. The drug-resistance mechanism provides a theoretical basis for the theoretical application of FTO inhibitors (e.g., Rhein, meclofenamic acid and FB23-2) to the treatment of GR NSCLC patients [114,115,116]. Meng et al. analyzed how m^6^A methylation affected afatinib resistance of NSCLC cells. They identified the m^6^A-modified genes, which might affect afatinib resistance by disturbing the cell cycle [117]. When NSCLC cells are in a state of oxidative stress, depletion of YTHDF1 suppressed translation of the Kelch-like ECH-associated protein 1 (Keap1) by recognizing the m^6^A modification and led to DDP resistance by activating the antioxidant reactive oxygen species clearance system [118].

For diagnosis, researchers built a diagnostic score model associated with 13 m^6^A regulators and concluded that the diagnostic scores of the LUAD patients were much higher compared with those of the control group [119]. This indicates that m^6^A modification could contribute the diagnosis of NSCLC patients. IGF2BP3 was highly expressed in LUAD tissue and led to worse prognosis of LUAD patients, which suggested that m^6^A modification had the potential to be used for predicting the prognosis of LUAD patients [120]. Huang et al. established an effective capture system for the preparation of the circulating tumor cells’ (CTCs) lysis, nucleic acid digestion and nucleoside extraction. They then applied liquid chromatography−electrospray ionization−tandem mass spectrometry (LC–ESI–MS/MS) to further analyze the m^6^A level of RNA in a single cell. This strategy provided a potential basis for determining whether metastasis occurs and assessing the prognosis of NSCLC patients [121].

## 5. Conclusions and Perspectives

The m^6^A modification of RNA, regulated by methyltransferases, demethylases and m^6^A-binding proteins, affects the target RNA splicing, translation, decay and nuclear export, which determines the development and progression of NSCLC. Increasing evidence has suggested that the effect of m^6^A modification on NSCLC patients’ prognosis is a double-edged sword, in that m^6^A modification has both promoting and inhibiting impacts on NSCLC development. As a prominent topic amongst epigenetic modifications, the roles of m^6^A modification of RNA in the progression and cellular biological functions of NSCLC are complicated. The reason for this complexity may be that m^6^A regulators have intricate effects on tumor-related RNA metabolism and that the inflammatory microenvironments and immune responses influence m^6^A modifications differently in different stagings and histological types of NSCLC. Besides, m^6^A modification acts as a critical factor that affects the treatment, drug resistance, diagnosis and prognosis of NSCLC patients.

RNAs with m^6^A modification may interact with signaling pathways or another regulatory network, such as the ceRNAs mechanism, to affect the progression of NSCLC and drug resistance of NSCLC patients, as in the description made by Jin et al. [70]. The existence of tRNA-derived small RNAs (tsRNAs) mutations in lung cancer samples indicates that tsRNAs can be a new diagnostic marker of lung cancer. It is believed that the diagnostic efficiency of m^6^A modification in NSCLC can be improved by combination with tsRNAs [122].

Targeted therapy and immunotherapy are still promising in the treatment of NSCLC; however, the corresponding drug resistance caused by m^6^A modification hampers their development. This reflects the importance of searching for inhibitors and activators of m^6^A regulators. Although m^6^A modification is not yet widely used in the clinical diagnosis and prognosis of NSCLC patients due to the complexity of detection, the RNA m^6^A level, to some extent, reflects whether lung cancer occurs and whether a NSCLC patient has a good prognosis. Therefore, further exploration of simpler and more feasible detection methods of RNA m^6^A level will be beneficial in the application of m^6^A modification in the diagnosis and prognosis of NSCLC patients. Gene-transduced adipocytes therapy was used for the treatment of breast cancer and was effective [123]. The inhibitors or activators of m^6^A regulators and tumor-related RNAs with or without m^6^A modification may be applied to the treatment of NSCLC patients via combination with Gene-transduced adipocytes therapy. The epidermal growth factor (EGFR) tyrosine kinase inhibitors (TKIs) has been effectively applied to the treatment of NSCLC by the combination of gefitinib with radiotherapy or chemoradiotherapy [124]. Further research into the application of EGFR TKs combined with m^6^A modification-related GR to the treatment of NSCLC patients may provide a new treatment program. Therefore, it is necessary to explore m^6^A RNA modification in the treatment and drug-resistance of NSCLC patients.

Although research into the effect of m^6^A modification of RNA on NSCLC is increasing, there are a number of problems that have yet to be solved, especially regarding the combined effect of m^6^A regulators on NSCLC. First of all, the gene expression levels of many m^6^A regulators in LUAD and LUSC tissues were significantly different to the levels in normal tissues [125]. Therefore, it is important to explore whether the other m^6^A regulators are associated with the development and progression of NSCLC. Secondly, because the m^6^A regulators are diversified and have various functions, different combinations determine the levels of RNAs m^6^A modification in NSCLC cells under certain conditions [125]. METTL3 and METTL14 both are methyltransferases, but their expression in LUAD tissues shows opposing tendencies; while FTO is a demethylase, METTL14 and FTO showed a similar expression trend [126]. Therefore, the intricate internal mechanism must be studied. Thirdly, some prognosis-associated mutant genes can impact the expression of m^6^A regulators in LUAD, such as TP53, which leads us to wonder whether those regulators are only effective in certain LUAD types of genetic mutations, if the regulators have different effects on LUAD development in different types of genetic mutation, or if there are other situations [104]. Finally, it has been proven that FTO and METTL3 are potential targets of diagnosis and therapy for LUAD patients [80,127]. In addition, the results of the diagnostic score model implied that YTHDF1 was expected to be a diagnostic marker and HNRNPC has the potential to act as a therapeutic target in LUAD [119], but further validation in LUAD populations is required.

## Figures and Tables

**Figure 1 genes-12-00440-f001:**
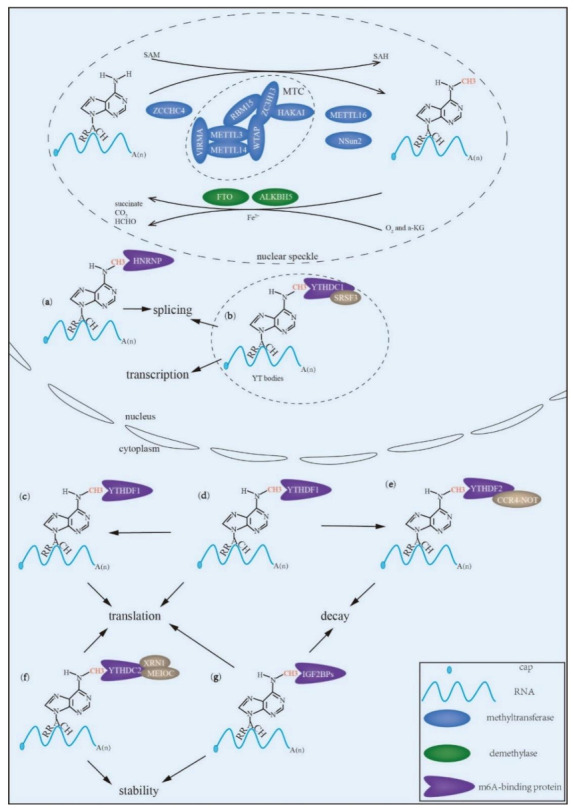
The regulation of RNA N6-methyladenosine (m^6^A) modification. The m^6^A modification of RNA is catalyzed by m^6^A methyltransferases including methyltransferase-like protein 3 (METTL3), METTL14, Wilms tumor 1-associated protein (WTAP), vir-like m^6^A methyltransferase associated (VIRMA), RNA-binding motif protein 15 (RBM15), zinc finger CCCH domain-containing protein 13 (ZC3H13), Cbl photo oncogene like 1 (CBLL-1, also termed HAKAI), NOP2/Sun domain family member 2 (NSun2) and CCHC zinc-finger-containing protein (ZCCHC4), and is removed by demethylases including fat mass and obesity-associated protein (FTO) and alkB homolog 5 (AlKBH5). In addition, the m^6^A modification is recognized and bound by m^6^A-binding proteins, which determine the fate of target RNAs—(**a**) Heterogeneous nuclear ribonucleoprotein (HNRNP) particles facilitate alternative splicing. (**b**) YT521-B homology domain-containing (YTHDC) 1 participates in alternative splicing, transcription and nuclear export. (**c**) YT521-B homology domain family (YTHDF) 1 enhances translation. (**d**) YTHDF3 regulates translation. YTHDF3 synergizes with YTHDF1 to modulate translation and cooperates with YTHDF2 to regulate decay. (**e**) YTHDF2 accelerates decay. (**f**) YTHDC2 participates in translation and increases stability. (**g**) Insulin-like growth factor 2 mRNA-binding proteins (IGF2BPs) augment translation, stability and decay.

**Table 1 genes-12-00440-t001:** Roles of m^6^A regulators in m^6^A RNA metabolism.

m^6^A Regulator	Roles in m^6^A RNA Metabolism	Reference
writers		
METTL3-METTL14	reinitializes transcription during UV-induced DNA damage responses	[16]
WTAP	influences RNA alternative splicing	[17]
METTL16	promotes mRNA splicing	[18]
RBM15	mediates mRNA degradation	[19]
erasers		
FTO	controls mRNA splicing	[20]
ALKBH5	impacts longer 3′-UTR mRNA splicing and stability	[21]
readers		
YTHDC1	participates in transcriptional processes, mRNA splicing and mRNA nuclear export, promotes the exon inclusion of mRNA	[22,23,24]
YTHDC2	participates in mRNA translation and mRNA stability	[25,26]
YTHDF1	enhances translational efficiency of mRNA	[27]
YTHDF2	regulates mRNA degradation, circRNA degradation and miRNA degradation	[28,29,30]
YTHDF3	regulates mRNA translation, circRNA translation and mRNAs degradation	[31,32,33]
IGF2BPs	prevent mRNA degradation, promote mRNA stability and alter lncRNA gene expression	[34,35]
HNRNP particles	affects mRNA abundance, mRNA alternative splicing, mRNA gene expression, RNA maturation of mRNA and RNA export pathway	[36,37]
EIF3	regulates mRNA translation	[38]

**Table 2 genes-12-00440-t002:** Dual effects of m^6^A mRNA modification on non-small cell lung cancer (NSCLC).

m^6^A Regulators	Role of m^6^A Modification in NSCLC	Function of m^6^A RNA in NSCLC	Reference
writers			
METTL3	oncogene	promotes cell EMT, migration and invasion	[70,71,72]
METTL14	anti-oncogene	unclear	[73]
KIAA1429	oncogene	promotes cell proliferation, invasion and migration	[74]
erasers			
ALKBH5	oncogene	promotes cell proliferation, invasion, migration and EMT	[75,76]
	anti-oncogene	inhibits cell under IH proliferation and invasion	[77,78]
FTO	anti-oncogene	inhibits cell proliferation and invasion	[79,80]
readers			
YTHDF3	oncogene	promotes cell growth, migration and invasion	[70]
YTHDF1	oncogene	promotes cell growth, EMT and invasion	[75]
YTHDF2	oncogene	promotes cell growth and proliferation	[81]
	anti-oncogene	inhibits cell growth, EMT and invasion	[75]
YTHDC2	anti-oncogene	unclear	[82]

**Table 3 genes-12-00440-t003:** Effects of m^6^A long noncoding RNAs (lncRNAs) modification on NSCLC.

m^6^A Regulators	Role of m^6^A Modification in NSCLC	Function of m^6^A RNA in NSCLC	Reference
writers			
METTL3	oncogene	promotes cell proliferation, EMT and brain metastasis	[70,83,84]
METTL3, METTL14	oncogene	unclear	[85]
readers			
YTHDF3	oncogene	unclear	[70]
YTHDF1, YTHDF2	oncogene	unclear	[85]

## Data Availability

Data sharing not applicable.

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
