# Peer review of "The Important Role of N6-methyladenosine RNA Modification in Non-Small Cell Lung Cancer"

_genes, 2021, doi:10.3390/genes12030440_

Round 1

Reviewer 1 Report

The manuscript by Yue Cheng et al is a comprehensive review on the role of N6-methyladenosine RNA modification in non-small-cell lung cancer. m6A modification has attracted much attention during recent years and is a prevalent modification found in mRNAs as well as several noncoding RNAs including tRNAs and rRNAs. Moreover, m6A modification is mediated by a large multi-subunit complex and the exerts its biological activity through specific recognition by several RNA binding proteins which are responsible for the regulation of gene expression during development as response to various cellular signals or stimuli. The authors attempt to focus on the role of m6A in NSCLC and to summarize various m6A regulators and their effect of m6A modification on cancer development, progression and cellular function. However, I feel that this is the weakest point of this review. Although the authors claim that this review is centered around NSCLC, it is actually not before page 7 (out of 11 of the main text) where the reader can find a correlation between m6A and NSCLC. The manuscript, although it’s not missing essential information, is poorly organized and requires extensive revision. The authors should move the last 4 pages upfront and restructure their manuscript in correlation to what is known about m6A and NSCLC and possibly other cancers. Table 1 although is informative is not exploited as the centerfold of this manuscript and the figure completely ignores important breakthrough studies regarding m6A and tRNA (or rRNA). Finally, the manuscript requires extensive language editing.  

Author Response

 Review(s)' Comments and Suggestions for Authors

Reviewer #1:

The manuscript by Yue Cheng et al is a comprehensive review on the role of N6-methyladenosine RNA modification in non-small-cell lung cancer. m6A modification has attracted much attention during recent years and is a prevalent modification found in mRNAs as well as several noncoding RNAs including tRNAs and rRNAs. Moreover, m6A modification is mediated by a large multi-subunit complex and the exerts its biological activity through specific recognition by several RNA binding proteins which are responsible for the regulation of gene expression during development as response to various cellular signals or stimuli. The authors attempt to focus on the role of m6A in NSCLC and to summarize various m6A regulators and their effect of m6A modification on cancer development, progression and cellular function.

Major revision:

  1. I feel that this is the weakest point of this review. Although the authors claim that this review is centered around NSCLC, it is actually not before page 7 (out of 11 of the main text) where the reader can find a correlation between m6A and NSCLC.

Answer:

First of all, we thank reviewer #1 very much for his/her critical review of our manuscript, his/her comments will help to improve the quality of our manuscript greatly. As your suggestion, we have simplified the well-known contents about m6A regulators including methyltransferases, demethylases and m6A-binding proteins to strengthen the significant role of m6A modification in NSCLC in our revise manuscript. For details, please check the revised manuscript with Track Changes.

  1. The manuscript, although it’s not missing essential information, is poorly organized and requires extensive revision. The authors should move the last 4 pages upfront and restructure their manuscript in correlation to what is known about m6A and NSCLC and possibly other cancers.

Answer:

Thank you for your valuable advice. We have reorganized the structure of our manuscript extensively including the title of each section. In addition, we add the section of ‘Roles of RNAs m6A Modification on Treatment, Drug Resistance, Diagnosis and Prognosis of NSCLC Patients’ in our revised manuscript. However, we think that the regulation of m6A modification to RNA metabolism is necessary for illustrating their roles in progression and cellular biological function as well as treatment and prognosis of NSCLC patients. Therefore, we have retained ‘The Regulation of RNAs m6A Modification’ in the original position of our revise manuscript.

  1. Table 1 although is informative is not exploited as the centerfold of this manuscript and the figure completely ignores important breakthrough studies regarding m6A and tRNA (or rRNA).

Answer:

Thank you for pointing out this issue. We have make the Table (Table1 has been divided into Table2 and Table3) be the centerfold of our manuscript. The posttranscriptional modification of tRNA (or rRNA) such as methylation has made breakthrough. For example, TRMT10A ablation leads to decreased N1-methylguanosine (m1G) in tRNA. N1-methyladenosine (m1A) at position 58 of tRNA is important to its stability. Nuclear-encoded enzyme TRMT2B catalyzes  5-methyluridine (m5U) in human mitochondrial tRNAs at position 54. More importantly, transcripts with increased m6A contribute the overrepresentation of m1G9-containing tRNAs codons read by tRNA Gln(TTG) , tRNA Arg(CCG) and tRNA Thr(CGT). While there was no corresponding article about m6A modification on tRNA (or rRNA) in NSCLC. Therefore, this content was not included in Table 3 of our revised manuscript.

  1. Finally, the manuscript requires extensive language editing.

Answer:

Thank you for your suggestion. We have done English language editing from MDPI. As a result, some changes or corrections including mistakes, typos, etc., were made. For details, please check the revised manuscript with Track Changes.

Lastly, we would like to thank reviewer#1 again for his/her excellent expertise and critical review of our manuscript.

Sincerely,

Hui Xu, M.D. & Ph.D. (on behalf of all authors)

Assistant Professor of Clinical Laboratory

The Cancer Hospital

Harbin Medical University

Tel: +86-139 3603 2383 (mobile)

Reviewer 2 Report

In the article entitled “The Important Role of N6-methyladenosine   RNAs  Modification in Non-small-cell Lung Cancer,” the authors prepared informative  review regarding the effect of  N6-methyladenosine  RNA    modification on cancer development, progression and cellular biological function in NSCLC.

The article is generally well written.  In preparing the review, the authors relied mainly on research results published over the past 3-5 years (2016-2021). The literature given in references  reflects the current level of knowledge in this issue.

However, there are some minor issues that I would like to draw attention to.  The text, for example, well describes how m6A-binding YTH  proteins work together to induce degradation of target m6A-mRNAs.  It is well known that m6A writers, erasers and readers frequently interact with each other, particularly with writers.

·         It would be more informative to provide data on cross-talk between pathways and co-operation of different regulators of m6A regulation of gene expression.

  • It would be also useful to show additionally in Figure 1 the chemical basis and molecular composition of the main regulators of m6A RNA methylation.
  • For a more complete analysis of the role of N6-methyladenosine modification and methylation (Lines 302-303) in the regulation of oncogenic and anti-oncogenic properties, Table 1 should be supplemented with fresh references to published research results and  discuss the reasons for the oncogenic and anti-oncogenic properties of m6A-mRNAs modifications.
  • In addition to long noncoding RNAs (Lines 388-407) it would be also very useful to present data and to give at least a brief discussion of the role of m6A modifications of other noncoding RNAs (microRNAs, circular RNAs) in function in NSCLC.

    In the article entitled “The Important Role of N6-methyladenosine   RNAs  Modification in Non-small-cell Lung Cancer,” the authors prepared informative  review regarding the effect of  N6-methyladenosine  RNA    modification on cancer development, progression and cellular biological function in NSCLC.

            •  

Author Response

Review(s)' Comments and Suggestions for Authors

Reviewer #2:

In the article entitled “The Important Role of N6-methyladenosine RNAs  Modification in Non-small-cell Lung Cancer,” the authors prepared informative  review regarding the effect of  N6-methyladenosine RNA modification on cancer development, progression and cellular biological function in NSCLC. The article is generally well written. In preparing the review, the authors relied mainly on research results published over the past 3-5 years (2016-2021). The literature given in references reflects the current level of knowledge in this issue.

Major revision:

  1. However, there are some minor issues that I would like to draw attention to. The text, for example, well describes how m6A-binding YTH proteins work together to induce degradation of target m6A-mRNAs.

Answer:

First of all, we thank reviewer#2 very much for his/her critical review and positive recognition of our work. We have added the sentence ‘Three YTHDF proteins work together to induce degradation of target m6A-mRNAs and one of them can fully or partially compensate for the effect of the others on target m6A-mRNAs degradation [106].’ in our revised manuscripts.

  1. It is well known that m6A writers, erasers and readers frequently interact with each other, particularly with writers. It would be more informative to provide data on cross-talk between pathways and co-operation of different regulators of m6A regulation of gene expression.

Answer:

Thank you for your good suggestion. We have provided data on cross-talk between pathways and co-operation of different regulators of m6A regulation of gene expression according to your suggestion in our revised manuscript. The sentences were as follows: ‘We can see that m6A modification is a dynamic, reversible and continuous process. Interaction between m6A regulators is common. For example, METTL3, METTL14 and WTAP compose a complex that promotes localization of MTC at nuclear speckles [51] and METTL3 interacts with eIF3h to enhance mRNA translation [50]. Therefore, we believe that m6A modification regulates RNA metabolism via the collective effect of m6A regulators.’

  1. It would be also useful to show additionally in Figure 1 the chemical basis and molecular composition of the main regulators of m6A RNA methylation.

Answer:

Thank you for your valuable advice. We have shown additionally in Figure 1 the chemical basis and molecular composition of the main regulators of m6A RNA methylation. For details, please check the Figure 1 of our revised manuscript.

  1. For a more complete analysis of the role of N6-methyladenosine modification and methylation (Lines 302-303) in the regulation of oncogenic and anti-oncogenic properties, Table 1 should be supplemented with fresh references to published research results and discuss the reasons for the oncogenic and anti-oncogenic properties of m6A-mRNAs modifications.

Answer:

Thank you for your valuable and thoughtful comments. We have added corresponding published research with two fresh references bellow in Table 1 (Table1 has been divided into Table 2 and Table 3): Deregulation of UBE2C-mediated autophagy repression aggravates NSCLC progression; N6-methyladenosine induced miR-143-3p promotes the brain metastasis of lung cancer via regulation of VASH1. We have discussed the reasons for the oncogenic and anti-oncogenic properties of m6A-mRNAs modifications as follows: m6A mRNA modification plays these different roles in the progression and cellular biological function of NSCLC likely due to different hepatic inflammatory microenvironments [115]. For details, please check the revised manuscript with Track Changes.

  1. In addition to long noncoding RNAs (Lines 388-407) it would be also very useful to present data and to give at least a brief discussion of the role of m6A modifications of other noncoding RNAs (microRNAs, circular RNAs) in function in NSCLC.

Answer:

Thanks for your excellent expertise in this aspect and important vision to deepen our work. We have presented data and given a brief discussion of the role of m6A modifications of microRNAs and circular RNAs in function in NSCLC (lines 461-483 of our revised manuscript). For details, please check the revised manuscript with Track Changes.

Lastly, we would like to thank reviewer 1 again for his/her excellent expertise and critical review of our manuscript.

Sincerely,

Hui Xu, M.D. & Ph.D. (on behalf of all authors)

Assistant Professor of Clinical Laboratory

The Cancer Hospital

Harbin Medical University

Tel: +86-139 3603 2383 (mobile)

Reviewer 3 Report

The manuscript entitled “The Important Role of N6-methyladenosine RNAs Modification in Non-small-cell Lung Cancer” authored by Cheng et al. deals with a topic of constantly increasing importance which is the epigenetic role in non-small cell lung cancer pathogenesis and development.

In the first part of review, the authors precisely presented the regulation of RNAs m6A modification by different factors and summarized all the information in a figure accompanied by a legend with sufficient information.

However, the second part, where the authors describe the presence of N6 methyladenosine in NSCLC, presents a large imbalance regarding the described information compared to the first part therefore, it requires major reconfiguration. Below please find my comments:

  1. In Introduction, authors mention that chemotherapy and surgery development has improved 5-year survival of NSCLC. This is true but targeted therapy first and immunotherapy afterwards were the real breakthroughs in NSCLC treatment. Please find more information here “Update 2020: Management of Non-Small Cell Lung Cancer, Alexander et al 2020”.

  1. Higher emphasis should be given in the included original studies. They need larger elaboration and presentation of more details for example, from this article: “Diagnostic, progressive and prognostic performance of m6A methylation RNA regulators in lung adenocarcinoma, Zhuang et al. 2020” since the manuscript has a biomedical context.

  1. Several interesting studies such as:

Deregulation of UBE2C-mediated autophagy repression aggravates NSCLC progression, Guo et al. 2018

Or

DOI: 10.1158/1541-7786.MCR-20-0541

DOI: 10.1089/dna.2020.6136

Which associate N6 methyladenosine modification with more (pre)clinical points of view could be easily added. Generally, I highly recommend to the authors to search relevant articles and include information regarding, association of this epigenetic event with treatment response, survival, prognosis etc. It is something that is mentioned in the conclusion but existing information from original studies has been overlooked by authors.

  1. A major revision of the English writing needs to be done. Expressions such as “what’s more”, “as we all know” should be rephrased in a more scientific written style.
  2. I propose to the authors to design a figure 2 like the figure 1 illustrating the mRNAs modified by the epigenetic regulators that have potential oncogenic or anti oncogenic role.
  3. In the conclusion, I would like to see more of the author’s personal points about this field of research. What inhibitors of this modulators exist today? Are there currently any clinical trials running with inhibitors of these epigenetic regulators? If no, do you suggest one? With what currently approved treatment could be combined?

Generally, the biomedical/(pre)clinical part of the manuscript should be significantly enriched.

Author Response

Review(s)' Comments and Suggestions for Authors

Reviewer #3:

The manuscript entitled “The Important Role of N6-methyladenosine RNAs Modification in Non-small-cell Lung Cancer” authored by Cheng et al. deals with a topic of constantly increasing importance which is the epigenetic role in non-small cell lung cancer pathogenesis and development. In the first part of review, the authors precisely presented the regulation of RNAs m6A modification by different factors and summarized all the information in a figure accompanied by a legend with sufficient information. However, the second part, where the authors describe the presence of N6 methyladenosine in NSCLC, presents a large imbalance regarding the described information compared to the first part therefore, it requires major reconfiguration. Below please find my comments:

Major revision:

  1. In the first part of review, the authors precisely presented the regulation of RNAs m6A modification by different factors and summarized all the information in a figure accompanied by a legend with sufficient information. However, the second part, where the authors describe the presence of N6 methyladenosine in NSCLC, presents a large imbalance regarding the described information compared to the first part therefore, it requires major reconfiguration.

Answer:

First of all, we thank reviewer #3 very much for his/her critical review of our manuscript, his/her comments will help to improve the quality of our manuscript greatly. As your suggestion, we have simplified the well-known contents about m6A regulators including methyltransferases, demethylases and m6A-binding proteins to strengthen the significant role of m6A modification in NSCLC in our revise manuscript. For details, please check the revised manuscript with Track Changes.

  1. In Introduction, authors mention that chemotherapy and surgery development has improved 5-year survival of NSCLC. This is true but targeted therapy first and immunotherapy afterwards were the real breakthroughs in NSCLC treatment. Please find more information here “Update 2020: Management of Non-Small Cell Lung Cancer, Alexander et al 2020”.

Answer:

Thank you for your keen eyes. We have cited this article and described the breakthrough of target therapy and immunotherapy in NSCLC treatment as follows: ‘Although targeted therapy and immunotherapy have made breakthroughs in NSCLC treatment [3], gene mutation and PD-1 expression remain obstacles in the treatment and the five-year survival rate of NSCLC patients is still unsatisfying [4].’

  1. Higher emphasis should be given in the included original studies. They need larger elaboration and presentation of more details for example, from this article: “Diagnostic, progressive and prognostic performance of m6A methylation RNA regulators in lung adenocarcinoma, Zhuang et al. 2020” since the manuscript has a biomedical context.

Answer:

Thank you for your valuable advice. We have elaborated more details about the original studies of the article above as follows: ‘For diagnosis, researchers built a diagnostic score model associated with 13 m6A regulators and concluded that diagnostic scores of the LUAD patients were much higher compared with that of the control group [165].’

  1. Several interesting studies such as: “Deregulation of UBE2C-mediated autophagy repression aggravates NSCLC progression, Guo et al. 2018” Or DOI: 10.1158/1541-7786.MCR-20-0541, DOI: 10.1089/dna.2020.6136, Which associate N6 methyladenosine modification with more (pre)clinical points of view could be easily added.

Answer:

Thank you for recommending these interesting studies associated with N6-methyladenosine modification with more (pre)clinical points of view. We have cited them in our revised manuscript as follows: ‘1.The m6A and expression level of mRNA UBE2C was decreased by ALKBH5. Downregulation of the mRNA activated the expression of the autophagy markers ATG3 and LC3B and reduced cell proliferation, clonogenicity, and invasive growth of NSCLC [122]. 2. Although the emergence of targeted therapy provides a new direction for the treatment of NSCLC, drug resistance has become a major obstacle during the treatment. Some research has indicated that m6A modification enhanced or attenuated drug resistance of NSCLC patients, such as resistance to gefitinib, afatinib and DDP. m6A modification decreased the gefitinib-resistance (GR) of NSCLC cells via the FTO/YTHDF2/ABCC10 axis [159]. 3. IGF2BP3 was highly expressed in LUAD tissue and led to worse prognosis of LUAD patients, which suggested that m6A modification had the potential to be used for predicting prognosis of LUAD patients [166].’

  1. Generally, I highly recommend to the authors to search relevant articles and include information regarding, association of this epigenetic event with treatment response, survival, prognosis etc. It is something that is mentioned in the conclusion but existing information from original studies has been overlooked by authors.

Answer:

Thanks for your excellent expertise in this aspect and important vision to deepen our work. We have added the section ‘Roles of RNAs m6A Modification on Treatment, Drug Resistance, Diagnosis and Prognosis of NSCLC Patients’ to describe the information regarding, association of RNAs m6A modification with treatment, prognosis etc. (lines 484-515 of our revised manuscript). For details, please check the revised manuscript with Track Changes.

  1. A major revision of the English writing needs to be done. Expressions such as “what’s more”, “as we all know” should be rephrased in a more scientific written style.

Answer:

Thank you very much for pointing out the language issues. We have revised corresponding expressions that are less scientific written style in our revised manuscript.

  1. I propose to the authors to design a figure 2 like the figure 1 illustrating the mRNAs modified by the epigenetic regulators that have potential oncogenic or anti oncogenic role.

Answer:

Thank you for providing this suggestion. We tried to design a figure 2 as you suggested, while we found the regulators are more complicate to put them in one figure. We would appreciate it if you could give me some advice about the Figure.  To make it more comprehensive, we reorganized Table 1 for illustrating the mRNAs modified by the epigenetic regulators that have potential oncogenic or anti oncogenic role.

  1. In the conclusion, I would like to see more of the author’s personal points about this field of research. What inhibitors of this modulators exist today? Are there currently any clinical trials running with inhibitors of these epigenetic regulators? If no, do you suggest one? With what currently approved treatment could be combined?

Answer:

Thanks for your excellent expertise in this aspect and important vision about our work. 1. The inhibitors of these modulators include Rhein, meclofenamic acid (MA) and FB23-2, which are FTO inhibitors. 2. There are not currently any clinical trials running with inhibitors of these epigenetic regulators. Although Rhein has the ability of inhibiting NSCLC cell proliferation, its water-soluble is poor, which limits the appliance in anti-tumor. Nonsteroidal anti-inflammatory drug MA specifically inhibits the demethylase activity of FTO and increases the overall m6A level in cells. However, lung adenocarcinoma cell is less sensitive to MA treatment, which limits its clinical application. FB23-2 can inhibit the demethylase activity of FTO and decrease the proliferative property of acute myeloid leukemia cells. There are currently no any clinical trials running with these FTO inhibitors. 3. Immunotherapy is currently approved treatment and could be combined with the inhibitors of these modulators. With the appliance of immunotherapy to anti-tumor, we think that these FTO inhibitors could be used for clinical trial combined with anti-inflammatory cytokine such as IL-37. For details, please check the forth and fifth chapter of our revised manuscript.

Lastly, we would like to thank reviewer#3 again for his/her excellent expertise and critical review of our manuscript.

Your favorable decision to our revised manuscript in the Journal of Frontiers in Genetics will be appreciated deeply.

With best wishes,

Sincerely yours,

Hui Xu, M.D. & Ph.D. (on behalf of all authors)

Assistant Professor of Clinical Laboratory

The Cancer Hospital

Harbin Medical University

Tel: +86-139 3603 2383 (mobile)

Reviewer 4 Report

The subject of this review is the role of N6-methyladenosine RNAs modification in NSCLC, however the text does not correspond with the topic.

In the manuscript, authors focused on the description of the regulation of RNAs m6A modification, rather than the role of this modification in lung cancer. At this point, this review does not differ from other articles in this topic. Moreover, authors focused on the characterization of proteins involved in the m6A modification, rather than its role and importance in RNA's metabolism, biological functions. The better way to present the list of m6A regulators (writers, erasers, readers) and their roles in RNA metabolism could be a table. It is worth to mention that the regulation of RNAs m6A modification is much better presented and described in Figure 1 than in the whole section 2 (with subchapters 2.1., 2.1.1., 2.1.2., 2.1.3., 2.1.4., 2.2., 2.2.1., 2.2.2., 2.3., 2.3.1., 2.3.2., 2.3.3. and 2.3.4.) discussed on 5 pages.

Authors should try to answer a few questions in their review, with avoidance of overall statements like "play an important role", "may be used", somehow, somewhere... How can m6A modification be used in the diagnosis of lung cancer? How can the knowledge about the regulatory mechanisms of m6A modification be included in the development of lung cancer therapies? What are the future perspectives of the role of m6A modification in lung cancer?

  • There is a typo error in the title of the 2.1.1 subsection (METTL3-METTL14).
  • Table 1 is an excellent data source, but its construction in the present form makes it a little bit incomprehensible.
  • Abbreviations should be checked and standardized at some point.
  • The abstract should be re-organized to define and emphasize the aim of this review. I suppose that summarization and presentation of the m6A regulators is not the aim of this review.
  • Authors should avoid colloquial speech. Instead, they should write briefly, say what they mean clearly, and avoid embellishment with unnecessary words.

Author Response

Review(s)' Comments and Suggestions for Authors

Reviewer #4:

The subject of this review is the role of N6-methyladenosine RNAs modification in NSCLC, however the text does not correspond with the topic.

Major revision:

  1. In the manuscript, authors focused on the description of the regulation of RNAs m6A modification, rather than the role of this modification in lung cancer. At this point, this review does not differ from other articles in this topic.

Answer:

First of all, we thank reviewer #4 very much for his/her critical review of our manuscript, his/her comments will help to improve the quality of our manuscript greatly. We have simplified the well-known contents about m6A regulators including methyltransferases, demethylases and m6A-binding proteins to strengthen the significant role of m6A modification in NSCLC in our revise manuscript. Also, we have reorganized the structure of our manuscript extensively including the title of each section. In addition, we add the section of ‘Roles of RNAs m6A Modification on Treatment, Drug Resistance, Diagnosis and Prognosis of NSCLC Patients’ in our revised manuscript. For details, please check the revised manuscript with Track Changes.

  1. Moreover, authors focused on the characterization of proteins involved in the m6A modification, rather than its role and importance in RNA's metabolism, biological functions. The better way to present the list of m6A regulators (writers, erasers, readers) and their roles in RNA metabolism could be a table.

Answer:

Thank you for providing this suggestion. We have removed some contents about characterization of proteins involved in the m6A modification and highlighted the key point of the role and importance of m6A modification in RNA metabolism, biological functions according to your suggestion. Besides, we have presented the list of m6A regulators (writers, erasers, readers) and their roles in m6A RNA metabolism into a table (Table 1 in our revised manuscript) according to your suggestion. For details, please check the revised manuscript with Track Changes.

  1. It is worth to mention that the regulation of RNAs m6A modification is much better presented and described in Figure 1 than in the whole section 2 (with subchapters 2.1., 2.1.1., 2.1.2., 2.1.3., 2.1.4., 2.2., 2.2.1., 2.2.2., 2.3., 2.3.1., 2.3.2., 2.3.3. and 2.3.4.) discussed on 5 pages.

Answer:

Thank you for your valuable advice. We have removed the subchapters according to your suggestion. In order to simplified Figure 1, we have supplemented the regulation of RNAs m6A modification in Table 1 of our revised manuscript.

  1. Authors should try to answer a few questions in their review, with avoidance of overall statements like "play an important role", "may be used", somehow, somewhere... How can m6A modification be used in the diagnosis of lung cancer? How can the knowledge about the regulatory mechanisms of m6A modification be included in the development of lung cancer therapies? What are the future perspectives of the role of m6A modification in lung cancer?

Answer:

Thank you for your valuable suggestion. We have replaced the overall statements like "play an important role", "may be used" into specific answer according to your suggestion in our revised manuscript. The response to the three questions above is as follows: 1. The construction of diagnostic score model associated with 13 m6A regulators contributes m6A modification to be used in the diagnosis of lung cancer. Besides, LC–ESI–MS/MS can be used for analyzing the m6A level of RNA in a single circulating tumor cell. This may provide a detective method to make m6A modification be used in the diagnosis of lung cancer. 2. The knowledge about the regulatory mechanisms of m6A modification be included in the development of lung cancer therapies. The mechanism of IL-37 used for treating NSCLC patients may be associated with m6A methylation. The drug resistance mechanism provides a theoretical basis for which the inhibitors of m6A demethylase FTO (e.g., Rhein, meclofenamic acid and FB23-2) are effectively applied to treatment of GR NSCLC patients. When NSCLC cells are in a state of oxidative stress, depletion of m6A reader YTHDF1 suppressed translation of Kelch-like ECH-associated protein 1 (Keap1) by recognizing the m6A modification and led to DDP resistance by activating the antioxidant reactive oxygen species clearance system. 3. The future perspectives of the role of m6A modification in lung cancer includes developing more m6A–related diagnostic markers and detective methods, exploring other RNAs modified by m6A in NSCLC, searching for the inhibitors and activators of m6A regulators (lines 522-553 of our revised manuscript). For details, please check the revised manuscript with Track Changes.

  1. There is a typo error in the title of the 2.1.1 subsection (METTL3-METTL14).

Answer:

Thank you for your careful work. We have corrected this error.

  1. Table 1 is an excellent data source, but its construction in the present form makes it a little bit incomprehensible.

Answer:

Thank you for pointing out this problem. We have restructured the Table 1 and separated the table into two and reconstruct the tables in terms of m6A regulators (Table 2, 3 of our revised manuscript) according to your suggestion.

  1. Abbreviations should be checked and standardized at some point.

Answer:

Thank you very much to point out this issue. We have checked and standardized the abbreviations.

  1. The abstract should be re-organized to define and emphasize the aim of this review. I suppose that summarization and presentation of the m6A regulators is not the aim of this review.

Answer:

Thank you for your suggestion on this issue. We have re-organized to define and emphasize the aim of this review, which is presenting the effect of m6A modification on cancer progression and cellular biological function of NSCLC as well as discussing how m6A modification affects treatment, drug resistance, diagnosis and prognosis of NSCLC patients.

  1. Authors should avoid colloquial speech. Instead, they should write briefly, say what they mean clearly, and avoid embellishment with unnecessary words.

Answer:

Thank you for your valuable recommendation. We have revised the corresponding colloquial speech and simplified the language of our manuscript and make it more accurate. For details, please check the revised manuscript with Track Changes.

Lastly, we would like to thank reviewer#4 again for his/her excellent expertise and critical review of our manuscript.

Your favorable decision to our revised manuscript in the Journal of Frontiers in Genetics will be appreciated deeply.

With best wishes,

Sincerely yours,

Hui Xu, M.D. & Ph.D. (on behalf of all authors)

Assistant Professor of Clinical Laboratory

The Cancer Hospital

Harbin Medical University

Tel: +86-139 3603 2383 (mobile)

Round 2

Reviewer 1 Report

The authors addressed all the points raised and the manuscript is now appropriate for publication.

Author Response

Review(s)' Comments and Suggestions for Authors

Reviewer #1:

The authors addressed all the points raised and the manuscript is now appropriate for publication.

Answer:

Thank you very much for your time and effort in reviewing and commenting our manuscript. We acknowledge your comments very much, which are valuable in improving the quality of our manuscript.

Sincerely,

Hui Xu, M.D. & Ph.D. (on behalf of all authors)

Assistant Professor of Clinical Laboratory

The Cancer Hospital

Harbin Medical University

Tel: +86-139 3603 2383 (mobile)

Reviewer 3 Report

The authors followed the instructions and improved sufficiently the manuscript. 

I can reconfirm that its quality has been improved significantly and meets the publication requirements of your journal.   I would like to highlight two points that I have noticed by reading the manuscript.  

  1. Line 333-335: "There is some connection between the application of EGFR TKs combined with gefitinib to the treatment of NSCLC and m6A modification-related GR".

The authors should rephrase because gefitinib is an EGFR TKI so probably the wanted to write some other type of therapy.        

2. Line 354: The sentence "further validation in LUAD populations is required" should be integrated in the text as complete sentence.  

Author Response

Review(s)' Comments and Suggestions for Authors

Reviewer #3:

The authors followed the instructions and improved sufficiently the manuscript. I can reconfirm that its quality has been improved significantly and meets the publication requirements of your journal. I would like to highlight two points that I have noticed by reading the manuscript.

Major revision:

  1. Line 333-335: "There is some connection between the application of EGFR TKs combined with gefitinib to the treatment of NSCLC and m6A modification-related GR". The authors should rephrase because gefitinib is an EGFR TKI so probably the wanted to write some other type of therapy.

Answer:

First of all, we thank reviewer 3# very much for his/her critical review and positive recognition of our work. As your suggestion, we have revised the sentence to ‘The further research about application of EGFR TKs combined with m6A modification-related GR to the treatment of NSCLC patients may provide a new treatment program.’ (lines 341-343 of our revised manuscript).

  1. Line 354: The sentence "further validation in LUAD populations is required" should be integrated in the text as complete sentence.

Answer:

Thank you for pointing out this issue. We have integrated the sentence "further validation in LUAD populations is required" in the text as a complete sentence (lines 365 of our revised manuscript). The sentence was changed to ‘In addition, the results of the diagnostic score model implied that YTHDF1 was expected to be a diagnostic marker and HNRNPC has the potential to act as a therapeutic target in LUAD [119], but further validation in LUAD populations is required.’ For details, please check the revised manuscript with Track Changes.

Sincerely,

Hui Xu, M.D. & Ph.D. (on behalf of all authors)

Assistant Professor of Clinical Laboratory

The Cancer Hospital

Harbin Medical University

Tel: +86-139 3603 2383 (mobile)

Reviewer 4 Report

Dear Authors,

I am pleased by the progress of Your manuscript.

However, maybe it is s just me, but I would adjust those phrases like "we can see," "we believe..." and replace it, e.g. "it can be seen," "it is believed that" oraz "there is some proof of that."

Kind regards

Author Response

Review(s)' Comments and Suggestions for Authors

Reviewer #4:

I am pleased by the progress of your manuscript.

Major revision:

  1. However, maybe it is just me, but I would adjust those phrases like "we can see," "we believe..." and replace it, e.g. "it can be seen," "it is believed that" or as "there is some proof of that."

Answer:

Thank you for your valuable advice. As your suggestion, we have revised the phrases "we can see," "we believe..." and replace it as "it can be seen", "it is believed that" in our revised manuscript. For details, please check the manuscript with Track Changes.

Sincerely,

Hui Xu, M.D. & Ph.D. (on behalf of all authors)

Assistant Professor of Clinical Laboratory

The Cancer Hospital

Harbin Medical University

Tel: +86-139 3603 2383 (mobile)